# Re-use of laboratory utensils reduces CO2 equivalent footprint and running costs

Martin Farley[1]*, Benoit P. Nicolet[2]*

**1** Sustainable UCL, University College London, London, United Kingdom, **2** Department of Hematopoiesis, Sanquin Blood Foundation & Landsteiner laboratory, Amsterdam UMC and Oncode Institute, Amsterdam, The Netherlands

☯ These authors contributed equally to this work.
¤ Current address: Department of Immuno-Oncology, The Netherlands Cancer Institute and Oncode Institute, Amsterdam, The Netherlands
* m.farley@ucl.ac.uk (MF); b.nicolet@nki.nl (BPN)

**Data Availability Statement:** All relevant data are within the manuscript and its Supporting Information files.

**Funding:** The author(s) received no specific funding for this work.

## Abstract

Laboratory-based research is resource intensive in terms of financial costs and its carbon footprint. Research laboratories require immense amounts of energy to power equipment, as well as large volumes of materials, particularly of single-use item consumption. In fact, many laboratories have essentially become reliant on single-use plastics. Understanding the full carbon footprint of consumable usage is increasingly important as many research institutes commit to carbon neutrality. To date, no carbon footprint assessment has been conducted to detail the differences between single-use plastics, and reusable glass in a laboratory setting. Here, we analyse the $CO_2$ equivalent ($CO_2$e) footprint of utilising single-use plastics, and re-use of glass or plastic items within laboratory environments. We focused our assessment on four commonly utilised consumables for mammalian cell and bacterial culture, and found that re-use scenarios resulted in substantial reduction in $CO_2$e footprint up to 11-fold. In addition, we estimated the long-term financial costs of re-use and single-use scenarios, and found that re-use had either similar or much lower running costs even when including technical staff wage. We concluded that research facilities must foster re-use in laboratory consumables, while reserving single-use items for select, defined cases. Our study highlights the need to account for indirect $CO_2$e footprint in designing a carbon-neutral lab and promotes circular economy principles.

## Introduction

Laboratory-based research is resource intensive in terms of financial costs [1], energy and material demand, which result in a high carbon footprint [2]. As research institutions establish carbon footprint reduction targets [3, 4], there is a need for quantifying the environmental impact of research and to identify mitigation strategies. To date, a limited number of $CO_2$ emission assessments from research intensive institutions have shown that the majority are due to scope 3 emissions [5, 6], which includes indirect emission of the making and disposal of materials used [7]. Many of such assessments though exclude scope 3 [8, 9], partly due to the

**Competing interests:** The authors have declared that no competing interests exist.

lack of existing carbon factors for goods and processes. As a result, the wider picture of laboratory emissions is incomplete.

Laboratories are distinct workspaces designed to handle chemical, physical and biological hazards. Yet, due to the nature of the potentially hazardous material handled in labs, the disposal of equipment and labware can present risks. This prevents in many cases the implementation of circular economy principles. Two key steps of the circular economy's closed-loop system rely on re-use or repurposing and recycling to minimize energy demand and maximize material efficiency [10]. In the lab environment however, these two key steps are practically difficult to implement. Indeed, re-use and recycling of materials requires energy or chemically intensive decontamination processes. The recycling of material coming from labs into consumer goods is even prevented by regulation in many countries to ensure biosafety. As breach of biosafety could have catastrophic impact on health and the planet's ecosystem, laboratory waste is often incinerated at high temperatures to effectively reduce the contamination risks. In many cases, these justifiably tough regulations for contaminated materials have led laboratories to adopt single-use labware.

Prior to the wide adoption of single-use plastics labware, glass utensils were used and re-used in laboratories, maximising material efficiency. Glassware, or derivatives thereof, confers chemical resistance to corrosive substances, is still found in most chemistry laboratories, and remains a vital resource within research spaces. Glassware and the reuse thereof, also comes with disadvantages: 1) decontamination and heavy cleaning are required for re-use and produce chemically contaminated waste waters; 2) technical assistance, associated salary, and equipment is needed to process items to re-use; 3) glass itself is often more expensive than plastic at the point of purchase driving up the costs of experimentation; 4) glass production is energy-intensive and requires mining work to extract the primary material [11]. As a result, laboratories often move away from glassware or re-use processes, fearing an increase in costs.

As glass has become less popular, scientists have become more concerned of the consumption and reliance on single-use plastics within laboratories [12]. Due to the use of fossil fuel as primary material for manufacturing, single-use plastics have become a centre of attention in the fight to tackle climate change [13]. Much of the single-use plastics in research laboratories, once utilised, are sent for high-temperature incineration. While incineration, in most cases, recovers part of the calorific value of single use equipment, it remains a carbon intensive and unsustainable process [14]. To mitigate these impacts, many labs have initiated efforts to improve the re-using [15] or closed-loop recycling of laboratory plastics [16]. Yet, time-demand of such practices has often prevented large scale adoption.

To date, the advantages and drawback of single-use of plastic and re-use of glass labware have not been quantified for the laboratory settings in terms of costs or carbon footprint. Studies from clinical environments have shown that reusable materials typically result in lower carbon footprints, and reduce financial costs [16, 17]. Yet, context is critical. For example, Life-cycle Assessments (LCAs) of bioreactors in a Good Manufacturing Practices (GMP) -compliant laboratories may not be transferable to non-GMP settings [18]. Equally, LCAs conducted in clinical settings are not be applicable in laboratory settings [19, 20], as these lack assessment of plastic consumables which are commonly utilised in laboratories.

To address this knowledge gap, we investigated how the $CO_2e$ footprint of utilising single-use plastics, or re-use of glass or plastic items within laboratory environments. We have focused our assessment on four commonly utilised single-use items for mammalian cell and bacterial culture: 6cm petri dishes, Pasteur pipettes, 50mL conical tubes, and 1L conical culture flask. We demonstrate that re-use scenarios are effective in reducing laboratory footprint and associated costs, even in long-term situations. Our work provides quantifiable evidences of

effective $CO_2e$ footprint reduction to laboratory managers and scientists, paving the way to a $CO_2e$ -efficient laboratory.

## Materials and methods

### Information sources for footprint calculations

For the assessment of the carbon dioxide equivalent ($CO_2e$) footprint of 6cm petri dishes, 50mL conical tubes, 1L conical flasks, and Pasteur pipettes, we searched Fisher Scientific's website (accessed at https://www.fishersci.co.uk, May to July 2021), a common supplier for specifications and costs of the consumable in plastic and a glass alternative. Values in $CO_2e$ for all the steps investigated in this study for each material was sourced from the 2020 UK government's conversion factors for company reporting of greenhouse gas emissions of the Department for Business, Energy & Industrial Strategy [11]. Cost of energy and water were taken from the author's local institutional energy / sustainability department for the year 2021.

### Usage scenarios

We designed use scenarios for single use plastics, re-used glass and re-use plastic item. We varied the number of items used from 1 item per weeks (52 per year) to 1000 items per week (52000 per year). For the re-use scenario, we assumed a re-use cycle of one week, thereby requiring only 1 week of item's supply. For each scenario, we calculated the footprint for production (extraction, primary processing, manufacturing and transporting materials to the point of sale), washing (electricity, water consumption and treatment), autoclaving (electricity, water consumption and treatment), drying (electricity), and disposal (incineration, with energy recovery). In the single-use plastic scenario, we assumed that all items were single-used and incinerated. For re-use scenario, for both glass and plastic items, we assumed a 10% breakage per year, for which we included the disposal footprint. Washing was assumed to take place in an energy efficient Miele laboratory glassware washer (PG8583) under standard cycle with no drying cycle employed. We contacted Miele for details and reported an electricity consumption of 1.5 kWh, and water consumption of 0.0435 $m^3$, under a standard cycle. Each tray within the unit was 47 x 52 cm, allowing to calculate the approximate number of items that could be fit within one wash. Autoclaving was assumed to take place in a direct steam generation Priorclave SH450 (Priorclave) with a water consumption of 0.402 $m^3$, and an electricity consumption of 64.375 kWh as reported previously [21]. The drying of items was assumed to take place in a fan-assisted high efficiency 425 Liter E3DWC425F/TDIG drying cabinet (Genlab), with a daily electricity consumption of 18.32 kWh. Of note, we do not endorse any of these products and solely used the values as purely indicative for our study. Additionally, in the case where items studied are not sufficient to fill an apparatus, we assume that the apparatus was not running half full and that other items were completing the run.

To allow for appropriate estimation of energy and water demand for washing we calculated the surface needed to wash 1 item and the number of items fitting in one washing tray, within a dishwasher that could fit 2 trays. We then adapted the load factor per run according to the number of items fitting per dishwasher. Similarly, to estimate the drying cabinet's load factor, we divided the item's surface (standing position) by the surface of 1 tray of the drying cabinet, with a maximum of 12 trays, as per manufacturer's information. For autoclaving, we calculated the volume occupied by one item and estimated the load factor by dividing the item's volume by the autoclave's internal volume. Salary estimates assumed a yearly salary of GBP 25,000 for a 7.2h workday with 5 weeks holidays. All analyses were conducted in Microsoft Excel (2016) and plotted in GraphPad (PRISM, version 9).

## Results

### Re-use of 50mL conical tubes reduces 11.3 times the $CO_2$e footprint compared to single-use tubes

We first estimated the $CO_2$e footprint resulting of the usage of 50mL conical tubes, either single-use in plastic, re-used glass alternative, or re-used plastic tubes (Fig 1A, S1A Fig). For reference, $CO_2$e is a defined as the carbon dioxide equivalent, which allows for more gases to be included alongside $CO_2$ and is the standard means of assessing carbon footprints. Single-use plastic tubes footprint includes the manufacture of raw material, the moulding, transport and disposal. For the reusable glass tube, the footprint includes the manufacture of raw material, the moulding, transport, washing, autoclaving, and drying. Of note, the footprint associated with washing included both the electricity and water treatment footprint. We have assumed that the equipment required for washing and drying of materials is already in place, and so have not included the embodied carbon of such equipment. We also have excluded packaging of single-use items. Water treatment $CO_2$e equivalent footprint also includes methane and $NO_x$ emission. Of note, in atmospheric chemistry, $NO_x$ is a generic term for the nitrogen oxide gases that are most relevant for air pollution. As glass maybe more susceptible to breakage, we accounted for a 10% per year breakage for the reusable glass tubes scenario, and included the footprint associated with their disposal. We assumed a weekly washing-

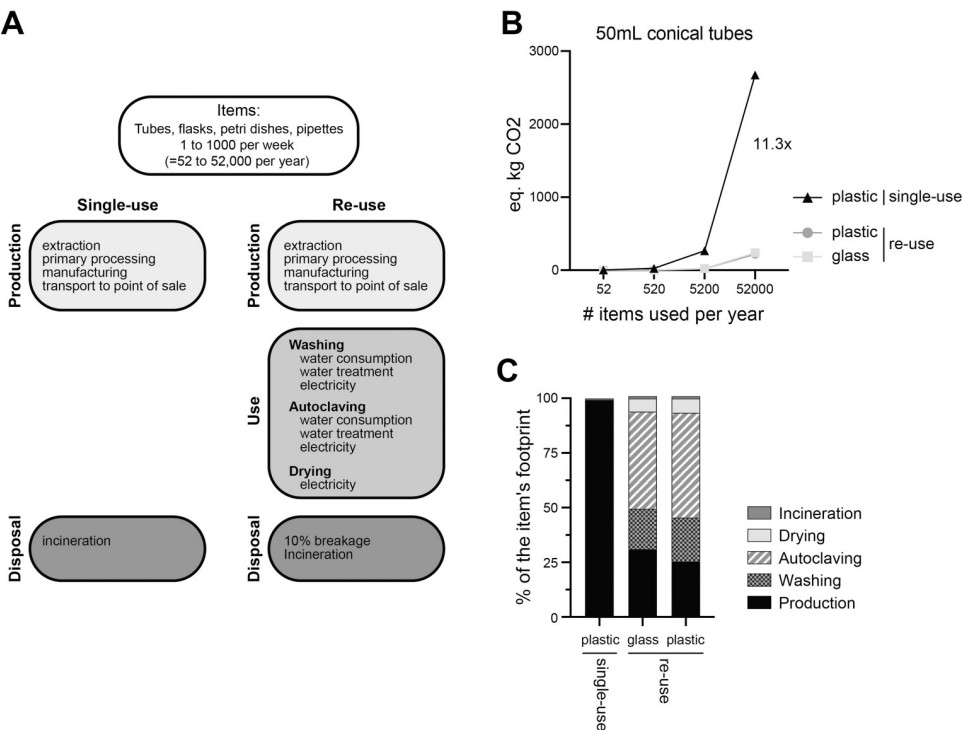

**Fig 1. $CO_2$ equivalent footprint of 50mL conical tubes.** (**A**) Representation of the system boundaries in the life cycle $CO_2$ equivalent ($CO_2$e) assessment, comparing single-use to re-used items commonly used in cell culture laboratories (50mL conical tubes, 6cm Petri dishes, Pasteur pipette and 1L conical flasks). For the re-use scenarios, a washing cycle of 1 week was assumed, requiring 52 times fewer items per year. 10% breakage was also assumed in the re-use scenarios. (**B**) $CO_2$e resulting from the use of single-use plastic or re-use of plastic or glass 50mL conical tubes. The number of items used varied from 52 per year (1 item reused every week) to 52,000 per year (1000 items reused every week). (**C**) Percentage of $CO_2$e per life cycle stages (production, washing, autoclaving, drying, and disposal) for the 3 scenarios in B.

autoclaving-drying cycle. Consequently, the estimated number of reusable glass tubes required per year is ~52 times lower than this of single use plastic tubes.

As research facilities do not use the same amount of tubes, we varied the usage scenarios from 1 item per week (52 items per year) to 1000 items per week (52,000 items per year). We found that independently of the number of items used per week, the single-use plastic tubes scenario generated 11.3 times more $CO_2$e equivalent than the reusable glass alternative (Fig 1B). The absolute difference between the single-use plastic and reusable glass tubes increased proportionally to the number of items used per year. We next considered the scenario where plastic tubes would be re-used as in the glass tubes scenario. We found that the re-used plastic tube scenario had a lower footprint than single-use plastics and even this of re-used glass tubes, albeit to a small extent (7.4% reduction compared to glass tubes; Fig 1B). We concluded that the re-use of tubes, whether glass or plastic, had a much lower $CO_2$e footprint than that of single-use.

We next examined the composition of the footprint for each scenario (Fig 1C). For this, we calculated the percentage of each scenarios' footprint, split by origin (production, washing, autoclaving, drying, incineration). We found that the single-use plastic tubes' footprint was almost exclusively due to the production with 99.3% of the total footprint, while incineration (with energy recovery) accounted for 0.67% of the footprint. The $CO_2$e footprint of the re-used scenario was more complex. For glass tubes, autoclaving accounted for 44%, production for 31%, washing 18.5%, drying 6%, and incineration of the 10% breakage accounted for 0.04% of the total footprint. For the re-used plastic tube scenario, the distribution was similar to glass, yet as the absolute production footprint was lower, it only accounted for 25.5% of the total footprint.

Altogether, we concluded that the $CO_2$e footprint associated with single-use plastic utilisation in laboratories is substantially greater than the re-use of either plastic or glass tubes. Notably, the footprint associated with production of single-use plastic tubes alone was responsible for the striking increase.

## Re-use of various labware has a lower $CO_2$e footprint than single-use plastic items

Labware comes in many different sizes and shapes. We hypothesised that the benefit of re-use may not be equal for all labware. Thereby, we reproduced our $CO_2$e footprint assessment for 3 other labware types: 6cm petri dishes, Pasteur pipettes, and 1L conical culture flasks. We found that irrespective of the type of labware, the re-use of items drastically reduced the $CO_2$e footprint compared to single-use items (Fig 2A–2C). Re-use of petri dishes had the lowest reduction with 2.8 times, while re-use of flasks had a 6.9 times reduction and that of Pasteur pipette a 10.6 times reduction in $CO_2$e footprint.

The origins of the $CO_2$e footprint for the 3 labware items were similar to conical tubes (Fig 1C). In all scenarios the $CO_2$e footprint of single-use plastics tubes was associated with production (~99% of the footprint), while incineration accounted for only ~1% of the footprint (Fig 2D–2F). In the re-use scenario, petri dishes and flasks had a similar partitioning of the footprint. Autoclaving accounted for most of the footprint with 51% for the petri dish scenario, while increasing to 69% in the flask scenario. Production and washing accounted for 10 to 15% of the $CO_2$e footprint while drying ranged from 4.4% (flask scenario) to 18.4% (petri dish scenario). In contrast, the re-use scenario for Pasteur pipettes revealed that the major part of the footprint was washing with 47.9% of the footprint followed by production with 35.8%. We concluded that re-use of labware reduced the $CO_2$e footprint compared to single-use items, where the differences was attributable to the production of single-use item.

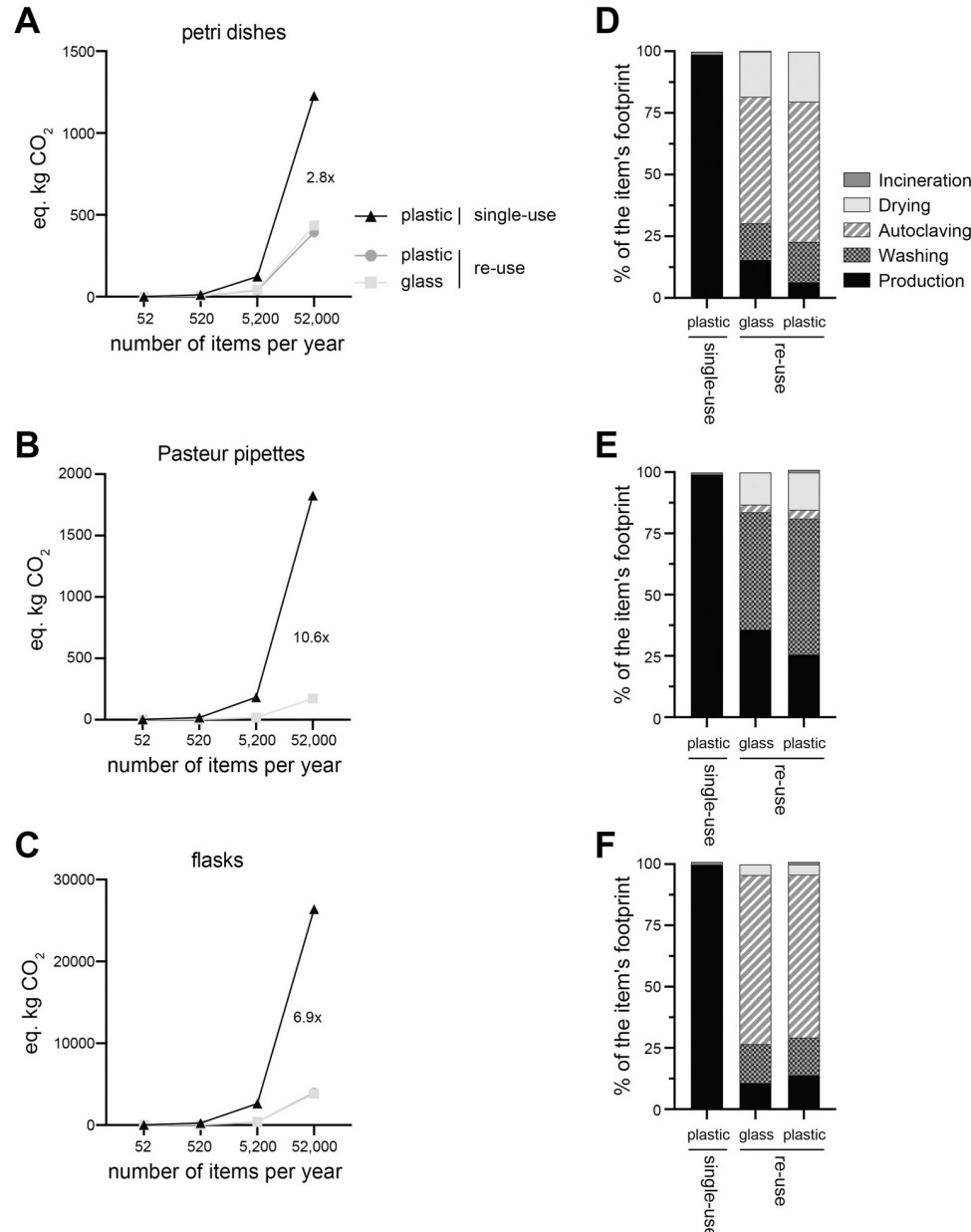

**Fig 2. CO$_2$ equivalent footprint of petri dishes, Pasteur pipettes, and conical flasks.** (**A-C**) CO$_2$e resulting from the use of single-use plastic or re-use of plastic or glass 6cm petri dishes, Pasteur pipette and 1L conical flasks. The number of items used varied from 52 per year (1 item reused every week) to 52,000 per year (1000 items reused every week). (**D-F**) Percentage of CO$_2$e per life cycle stages (production, washing, autoclaving, drying, and disposal) for the 3 scenarios in A-C.

## Cost of re-use is similar or lower than this of single use plastic items

Finally, we investigated the cost estimates for single-use or re-use scenarios. As running costs are never fully captured after a year of use, we calculated the costs over a period of 10 years of use (S1B Fig). We first examined the cost estimate of conical tubes in absence of salary costs for technical handling. Glass tubes were more expensive at purchase per item than plastic tubes (S1 Table). Yet over the course of 10 years, re-use of glass tubes leads to a decrease of

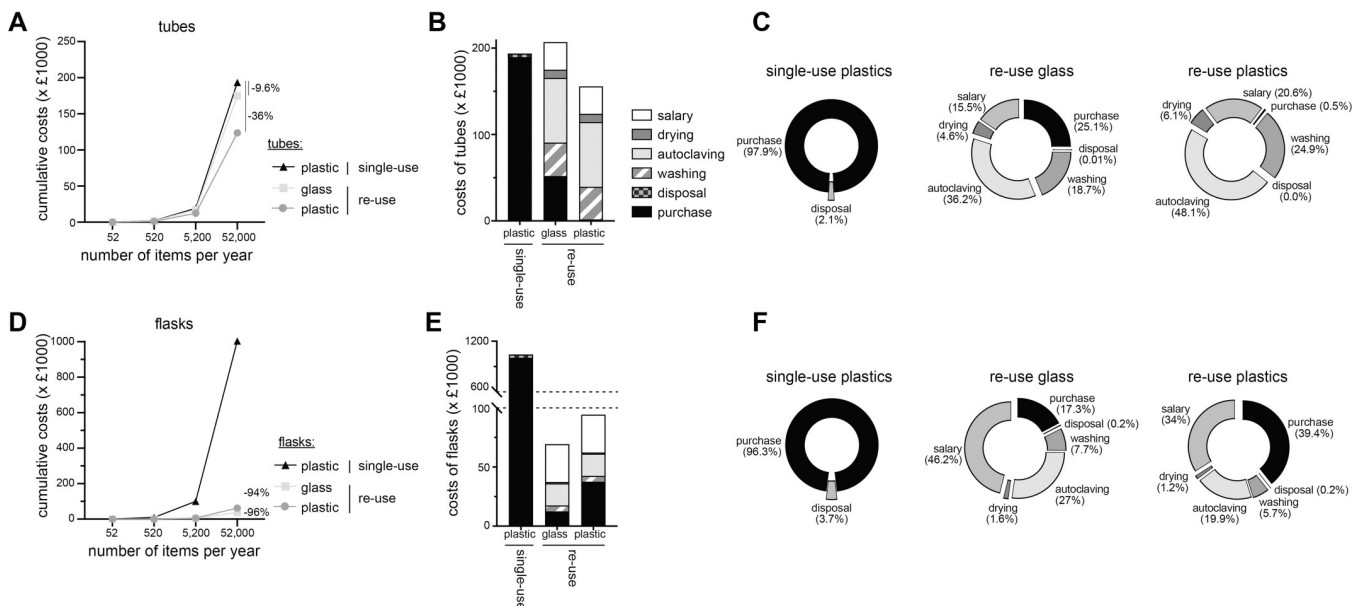

**Fig 3. Costs associated with single-use and re-use of tubes and flasks for 10 years.** (**A**) Running costs of a 10-year period for 50mL conical tubes excluding costs associated with salary of support staff for re-use scenario. (**B-C**) Costs of tube per life cycle stage (production, washing, autoclaving, drying, and disposal) including salary for technical staff (B) and percentage of total costs (C) for the 3 scenarios of A. (**D-F**) similar to A-C for 1L conical flasks.

9.6% of the running cost compared to single-use items (Fig 3A). Re-use of plastics tubes had a cost estimate reduction of 36%, due to the cheap price per item. We concluded that cost estimates of re-use scenario are lower than that of single-use plastics over the course of 10 years.

We next added the cost estimates of salary for staff involved with washing, autoclaving and drying tubes. To better approximate the share of salary in the running costs, we estimated a hands-on time of 15s per items (S1 Table). Of note, the hands-on time per item will decrease with larger number of items, which was not accounted for in our calculation. We found that adding the salary estimates onto the running cost of glass tube re-use, led to a 6% increase of cost estimates over this of single use plastics (Fig 3B, S2A Fig). Plastics tube re-use scenario still resulted in a 19.5% decrease in cost estimate compared to single-use plastic tube scenario (Fig 3B, S2A Fig). Taken together, the costs associated with re-use of glass or plastic tubes were either slightly higher or much lower than these of single-use plastic tubes.

We next examined the costs per life cycle stage (production, washing, autoclaving, drying, and disposal) for tubes (Fig 3C, S2B Fig). We found that 97.9% of the cost of single-use plastic tubes was attributed to the purchase and only 2.1% was associated with disposal. Re-use scenarios were highly influenced by the purchase costs. Indeed, purchase costs of glass tubes were higher than these of plastic tubes and represented 25.1% of the total costs compared to 0.5% for plastic tubes (Fig 3C). Costs associated with autoclaving had the largest share of costs with 36.2% and 48.1%, for glass and plastics respectively. Cost of washing (glass: 18.7%, plastic: 24.9%), salary of technical staff (glass: 15.5%, plastic: 20.6%), drying (glass: 4.6%, plastic: 6.1%) and disposal (glass: 0.01%, plastic: 0%) where similar in both re-use scenarios. Together, costs associated with re-use of tubes were similar or lower than those of single-use tubes.

We next conducted a similar analysis of running cost of 1L conical culture flasks. We again found dramatic reduction in cost estimates when reusing flasks (94% to 96%), compared to single-use plastics flasks (Fig 3D). Similar findings were observed when including salary estimates in the total costs (plastic re-use: -91%; glass re-use: -93%; Fig 3E, S2C Fig). The distribution of running costs of single-use flasks were similar to these of conical tubes (purchase:

96.3%; disposal 3.7% of total costs; Fig 3F, S2C Fig). Re-use scenarios were again highly influenced by the purchase costs, where purchase of plastic flasks represented 39.4% of the total costs compared to 17.3% for glass flasks (Fig 3F). Running costs distribution of re-use scenario of glass and plastic were similar: salary (glass: 46.2%, plastic: 34%), autoclaving (glass: 27%, plastic: 19.9%), washing (glass: 7.7%, plastic: 5.7%), drying (glass: 1.6%, plastic: 1.2%) and disposal (0.2% for both). We concluded that in the case of flasks, the running costs of re-use scenario were substantially lower compared to single-use scenario.

Altogether, we conclude that costs of reusing labware range from slightly higher (+6%), to much lower (-93%) costs than this of single-use plastics. This cost reduction was even more striking in the case of re-using plastics flasks.

## Discussion

In this study we evidence that re-use of items, as opposed to single-use, leads to substantially lower carbon footprints, irrespective of whether one uses glass or plastic. This was true for all four types of labware assessed. The main differences between re-use and single-use scenarios resided in the number of items needed to be produced, particularly when there was high usage. We would like to stress that plastic, as a material, is not the factor that influences its carbon footprint compared to glass items. Indeed, the key factor is the single-use nature of plastic. Single-use items require *de novo* production of items which represents ~99% of the footprint. In contrast, re-using items reduced the need for *de novo* production of items, thereby reducing the overall carbon footprint.

When re-used, labware items require energy to clean, wash, and dry. Yet re-use had a lower carbon footprint than production of new items required by single-use scenario. We expect this difference to increase as carbon emissions associated with energy production are decreasing in many countries due to the increase in renewable sources of energy [22, 23]. Single-use items, in contrast, are typically incinerated at high temperatures. While incineration resulted in large carbon emissions, it was negligeable in comparison to those associated with the production of new items. In fact, our assessment showed that the impacts of waste processes were consistently lower than treatment or production, whether recycled or incinerated. Of note, for the calculation of emission related to incineration, we assumed that items did not contain any liquid. Incinerating items containing liquids or remaining organic compounds will significantly further increase the footprint of single-use items' incineration.

Not all laboratory items may be suitable for re-use. For example, some plastics do not withstand heat from autoclaves. For the purpose of this study, we selected common consumables which can be utilised in high volumes, and have common reusable alternatives readily available. Items which are composed of multiple materials, or are not easily separated and washed (i.e. cellulose pore filters), may not be appropriate for re-use. Furthermore, contamination in some cases may pose great risk, and re-use may not be feasible (e.g. in the clinical setting). Scientists must have a degree of certainty for crucial experiments that some items are not contaminated. Otherwise there exists the risk of having to duplicate an experiment, which would negate any positive impacts of re-using items. Historically, and still in many parts of the world, science facilities have relied on local re-use of common consumables. In recognition of the requirement of some unavoidable single-use items, facilities should seek a balance between cost, environmental impact, and contamination. We recommend to re-use where feasible, but have single-use available for limited agreed uses.

Determining the $CO_2e$ required for the manufacturing of consumables was based on standardised national (UK) figures associated with plastic and glass production. Transparent reporting on embodied carbon of manufacturing consumables would permit purchasers of

consumables to better compare outcomes. One additional limitation of our study is the conversion of all impacts into $CO_2$ equivalence. Re-use of materials on-site requires not just water and energy, but also cleaning products and detergents for which $CO_2$ equivalence may not capture the overall footprint. LCAs for detergents thus far have been limited to household applications [24], making their use for our study of laboratory spaces challenging. Some of these impacts may be mitigated through filtering and treatment of outgoing water supply [25]. Future studies should include life-cycle assessments of the footprint of laboratory detergents.

When institutions set "Net-zero" emission targets, the definition of net-zero can change drastically. Carbon emissions are divided into 3 levels, Scope 1, 2, and 3. Net-zero emission targets will often focus on scopes 1 and 2, which will often assess energy consumption and water use (which are typically required for washing reusable materials). Scope 3 emissions include the embodied carbon required to produce materials, such as single-use plastics. As a result, research institutions which set net-zero targets only for scopes 1 and 2 emissions may be inadvertently resulting in increased overall emissions, by increasing usage of single-use items to avoid use of energy intensive washing equipment.

Design of laboratories presents this challenge in balancing all scopes of carbon emissions, as architects may seek to minimise the sizes of autoclaves and washing facility as a way of carbon emissions reduction. Our findings indicate that the exact opposite should happen: by minimising the capacity of a research facility to decontaminate, wash, and re-use materials, facilities are in fact increasing their overall carbon footprint. We also show that facilities are likely to increase overall costs as well. New facilities should determine what long-term capacity may be required for on-site re-use, and design backwards from there. Notably, the environmental impact of cleaning and washing on-site has been decreasing due to reduction of $CO_2$e emissions associated with energy production, rendering re-use even more favourable than single-use.

Many labs are not pressed for their impact in terms of carbon footprint, but rather are under significant financial restraints. Financial pressures, along with supply issues, can force laboratories to implement re-use strategies, indicating a likelihood that re-use is cheaper. Lower price of re-used items was also apparent in our findings: re-use strategies were either similar, or much cheaper over time. This remained so even when factoring in wages for support staff to conduct the washing. Note that our study only assessed the impacts by looking at individual consumables. The overall savings of a central wash facility would scale up as more items are being re-used. Institutional financial pressures can also be responsible for the undersizing of wash facilities, as larger facilities typically require central funding as opposed to that of a single laboratory. As a result, lack of central funding and upkeep will limit the capacity for on-site re-use.

Our study highlights the need to implement re-use strategies, regardless of the material use in laboratory consumables. These re-use strategies will be crucial if research processes are to approach net-zero emissions. While there are limited guidance materials, scientists should consult resources such as the Sustainable Consumables Guide from UCL's LEAF programme [26], and consider how such circular economy principles may be applied locally. Our findings also indicate the need for systematic carbon footprint assessment, including all three scopes of carbon emissions, in contrast to institute-centric assessments. Reusing laboratory consumables not only increases the share of funding for actual research, but also paves the way to carbon-neutral institutes.

## Supporting information

**S1 Fig. Graphical representation of the methodology used in this paper.** (A-B) Representation of the calculation methodology for (A) $CO_2$ equivalent and (B) costs

(see Materials and methods).
(JPG)

**S2 Fig. Costs associated with single-use and re-use of tubes and flasks for 10 years including salary of support staff.** (**A-C**) Running costs of a 10-year period for (A-B) 50mL conical tubes and (B-C) conical 1L flasks including costs associated with salary of support staff for re-use scenario. (B) Presents the details of (A,C) in x1000 GBP for the scenario of 52 000 items used per year.
(JPG)

**S1 Table. $CO_2e$ calculation of labware.** This table details the calculation used in this study.
(XLSX)

## Acknowledgments

We would like to thank Marta Rodriguez-Martinez, Branka Popovic, and Monika Wolkers for critical reading of this paper.

## Author Contributions

**Conceptualization:** Martin Farley, Benoit P. Nicolet.

**Data curation:** Martin Farley, Benoit P. Nicolet.

**Formal analysis:** Martin Farley, Benoit P. Nicolet.

**Investigation:** Martin Farley, Benoit P. Nicolet.

**Methodology:** Martin Farley, Benoit P. Nicolet.

**Project administration:** Martin Farley, Benoit P. Nicolet.

**Resources:** Martin Farley, Benoit P. Nicolet.

**Software:** Benoit P. Nicolet.

**Supervision:** Martin Farley, Benoit P. Nicolet.

**Validation:** Martin Farley, Benoit P. Nicolet.

**Visualization:** Benoit P. Nicolet.

**Writing – original draft:** Martin Farley, Benoit P. Nicolet.

**Writing – review & editing:** Martin Farley, Benoit P. Nicolet.

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
