## [Decision Letter · Decision Letter 0]

8 Nov 2022

PONE-D-22-26188Re-use of labware reduces CO2 equivalent footprint and running costs in laboratoriesPLOS ONE

Dear Dr. Farley,

Thank you for submitting your manuscript to PLOS ONE. After careful consideration, we feel that it has merit but does not fully meet PLOS ONE’s publication criteria as it currently stands. Therefore, we invite you to submit a revised version of the manuscript that addresses the points raised during the review process.

We look forward to receiving your revised manuscript.

Kind regards,

Atif Jahanger, Ph.D

Academic Editor

PLOS ONE

Journal Requirements:

Reviewers' comments:

Reviewer's Responses to Questions

**Comments to the Author**

1. Is the manuscript technically sound, and do the data support the conclusions?

Reviewer #1: Yes

Reviewer #2: Yes

2. Has the statistical analysis been performed appropriately and rigorously? 

Reviewer #1: N/A

Reviewer #2: Yes

3. Have the authors made all data underlying the findings in their manuscript fully available?

Reviewer #1: Yes

Reviewer #2: Yes

4. Is the manuscript presented in an intelligible fashion and written in standard English?

Reviewer #1: Yes

Reviewer #2: Yes

5. Review Comments to the Author

Reviewer #1: The authors have attempted to test the effect of a concept related to circular economy, which is truly a time-demanding topic. The overall work is good. However, I feel that there is still some room to improve the paper before it can be published. My suggestions are attached to a separate file.

Reviewer #2: The study titled "Re-use of labware reduces CO2 equivalent footprint and running costs in laboratories"

First, the title should be slightly improved.

The abtract must be more concrete to explicitly mention the real contribution of the paper.

A slight improvement in the information flow will add worth to the manuscript.

Lastly, a suggested reading from other disciplines to make literature citation more strong and diverse; https://doi.org/10.1007/s11356-022-20320-z

6. PLOS authors have the option to publish the peer review history of their article (what does this mean?). If published, this will include your full peer review and any attached files.

Reviewer #1: No

Reviewer #2: **Yes: **Ashar Awan

---

## [Author Response · Author response to Decision Letter 0]

23 Jan 2023

Reviewers' comments: 

Reviewer #1: 

The authors have attempted to test the effect of a concept related to circular economy, which is truly a time-demanding topic. The overall work is good. However, I feel that there is still some room to improve the paper before it can be published. My suggestions are attached to a separate file. 

(Authors pasted below the separate file provided by the reviewer, for convenience)

Abstract: 

1. The abstract is well-written, clarifying the reasons of this study and highlighting the main findings. 

Thank you.

Introduction 

1. Please include page numbers and/or line numbers. 

Thank you, we have now added page numbers to the manuscript (as also required for manuscript submission). 

2. The background information is insufficient and needs to be elaborated. The authors should be able to build up a concrete and coherent relationship between the circular economy of labware use, its effect on CO2 emissions and other environmental benefits along with the economic gains of the reuse. And then the research gaps need to unfolded step by step. Please include one or two paragraphs detailing this information.

We thank the reviewer for asking more detailed and coherent introduction. We have now entirely re-writen the introduction to: 1) better highlight the knowledge gap and how we address it in this study; 2) better introduce our work in the context of the circular economy; 3) highlight the clear difference of the laboratory resource and waste management compared to classical household setting (e.g. the use of biohazard material prevents the recycling of material in many countries). 

3. The novelty statement is missing and the claimed novelties have been posted without any detailed literature review. I suggest writing the originalities of this article point-by-point accounting the recent progress in the literature. This is section where the authors need to show how they stand out from other existing papers in the scholarship.

We thank the reviewer for asking for clarification of the novelty aspect of our work. Specifically, as mentioned in our answer to comment #2 of the reviewer, we have re-written the introduction to better highlight the problematic addressed and the novelty of our work. In addition, we would like to emphasize that to our knowledge there are no studies regarding the quantification of CO2e footprint of labware in the context of the laboratory environment (again, as indicated in the new introduction, is very different from the household environment due to biohazardous materials). 

4. I do not see any separate literature review section. Also, the introduction section does not dive deep into the existing literature.

We thank the reviewer for this comment. As mentioned in our answer to the comment #3 of the reviewer, there is to our best knowledge no prior literature on this subject. This prevents us from providing any depth in a field which is understudied. 

5. Please number the sections and sub-sections accordingly.

We thank the reviewer for this comment. However, we refrain from numbering our sections due to journal’s formatting guidelines. Nonetheless, we now included a level 1 and 2 heading style, as per journal guideline, to clarify the sectioning structure of our article. 

Methods

1. The method section looks very abstract. It would be great if the authors could include a pictorial flowchart detailing the methodological framework used in this paper. 

We thank the reviewer for this nice suggestion. We now include a flow-chart to graphically detail the methodology used in this paper. 

Empirical Analysis and Conclusion 

The results and discussion sections are well-written. However, the conclusion and policy implications are missing. Please include these and specify the limitations of this study. Along with that suggest for the future studies. 

We thank the reviewer for this comment. The new version of our discussion addresses the concerns of the reviewer. We believe that we highlight the policy implication, provided further context into future laboratory design, and provided suggestions for future studies to improve upon our own. 

Reviewer #2: 

The study titled "Re-use of labware reduces CO2 equivalent footprint and running costs in laboratories" 

First, the title should be slightly improved. 

The abtract must be more concrete to explicitly mention the real contribution of the paper.

A slight improvement in the information flow will add worth to the manuscript. 

Lastly, a suggested reading from other disciplines to make literature citation more strong and diverse; https://doi.org/10.1007/s11356-022-20320-z

We thank Dr. Ashar Awan for the suggestions to improve our manuscript. The title was changed to “Re-use of laboratory utensils reduces CO2 equivalent footprint and running costs” to better reflect the findings of our paper. We now entirely rearranged the introduction to highlight the paper’s problematic, and the knowledge gap which we address. The information flow was also reworked. We now implement the suggestion of reviewer #1 to include a flow chart to help the reader understand our study design and workflow (see new Fig S1). Finally, we thank the reviewer for the reading suggestion. It is a nice article. To make the citation more diverse, we also included several new citations (see references list).

---

## [Decision Letter · Decision Letter 1]

14 Mar 2023

Re-use of laboratory utensils reduces CO2 equivalent footprint and running costs

PONE-D-22-26188R1

Dear Dr. Farley,

We’re pleased to inform you that your manuscript has been judged scientifically suitable for publication and will be formally accepted for publication once it meets all outstanding technical requirements.

Kind regards,

Atif Jahanger, Ph.D

Academic Editor

PLOS ONE

Additional Editor Comments (optional):

Reviewers' comments:

Reviewer's Responses to Questions

**Comments to the Author**

1. If the authors have adequately addressed your comments raised in a previous round of review and you feel that this manuscript is now acceptable for publication, you may indicate that here to bypass the “Comments to the Author” section, enter your conflict of interest statement in the “Confidential to Editor” section, and submit your "Accept" recommendation.

Reviewer #1: All comments have been addressed

Reviewer #2: (No Response)

2. Is the manuscript technically sound, and do the data support the conclusions?

Reviewer #1: Yes

Reviewer #2: (No Response)

3. Has the statistical analysis been performed appropriately and rigorously? 

Reviewer #1: Yes

Reviewer #2: (No Response)

4. Have the authors made all data underlying the findings in their manuscript fully available?

Reviewer #1: Yes

Reviewer #2: (No Response)

5. Is the manuscript presented in an intelligible fashion and written in standard English?

Reviewer #1: Yes

Reviewer #2: (No Response)

6. Review Comments to the Author

Reviewer #1: The authors have addressed all of my comments. This version reads well and can be accepted for publication.

Reviewer #2: (No Response)

7. PLOS authors have the option to publish the peer review history of their article (what does this mean?). If published, this will include your full peer review and any attached files.

Reviewer #1: No

Reviewer #2: **Yes: **ashar awan

---

## [Editor Report · Acceptance letter]

21 Mar 2023

PONE-D-22-26188R1 

Re-use of laboratory utensils reduces CO2 equivalent footprint and running costs 

Dear Dr. Farley:

I'm pleased to inform you that your manuscript has been deemed suitable for publication in PLOS ONE. Congratulations! Your manuscript is now with our production department. 

Kind regards, 

on behalf of

Dr. Atif Jahanger 

Academic Editor

PLOS ONE